# Design and characterization of hyperactive mutants of the *Agrobacterium tumefaciens* telomere resolvase, TelA

**Shu Hui Huang, Kayla Abrametz, Siobhan L. McGrath¤, Kerri Kobryn** *

Department of Biochemistry, Microbiology & Immunology, College of Medicine, University of Saskatchewan, Saskatoon, Saskatchewan, Canada

¤ Current address: Global Institute of Food Security, Saskatoon, SK, Canada
* kerri.kobryn@usask.ca

**Data Availability Statement:** All relevant data are within the manuscript and its Supporting Information files.

## Abstract

Telomere resolvases are a family of DNA cleavage and rejoining enzymes that produce linear DNAs terminated by hairpin telomeres from replicated intermediates in bacteria that possess linear replicons. The telomere resolvase of *Agrobacterium tumefaciens*, TelA, has been examined at the structural and biochemical level. The N-terminal domain of TelA, while not required for telomere resolution, has been demonstrated to play an autoinhibitory role in telomere resolution, conferring divalent metal responsiveness on the reaction. The N-terminal domain also inhibits the competing reactions of hp telomere fusion and recombination between replicated telomere junctions. Due to the absence of the N-terminal domain from TelA/DNA co-crystal structures we produced an AlphaFold model of a TelA monomer. The AlphaFold model suggested the presence of two inhibitory interfaces; one between the N-terminal domain and the catalytic domain and a second interface between the C-terminal helix and the N-core domain of the protein. We produced mutant TelA's designed to weaken these putative interfaces to test the validity of the modeled interfaces. While our analysis did not bear out the details of the predicted interfaces the model was, nonetheless, extremely useful in guiding design of mutations that, when combined, demonstrated an additive activation of TelA exceeding 250-fold. For some of these hyperactive mutants stimulation of telomere resolution has also been accompanied by activation of competing reactions. However, we have also characterized hyperactive TelA mutants that retain enough autoinhibition to suppress the competing reactions.

## Introduction

Spirochetes of the *Borrelia* genus and several species of Agrobacterium possess linear replicons terminated by covalently closed hairpin (hp) telomeres. The replication cycle of these linear DNAs is thought to involve initiation of replication at an internal origin that sends out a replication fork towards each hp telomere [1–3]. As replication is completed the hp telomeres are converted into replicated telomere junctions (*rTel*) present in an inverted repeat circular

**Funding:** KK was supported by Discovery Grants from the Natural Sciences and Engineering Research Council of Canada (NSERC; RGPIN 04382-2017, GRF-0-2006, RGPIN-2024-05101) and by a CoMRAD grant from the University of Saskatchewan's College of Medicine (2022-2023). There was no additional external funding received for this study.

**Competing interests:** The authors have declared that no competing interests exist.

dimer. This dimer is resolved into a pair of linear DNAs terminated by the hp telomeres by a DNA cleavage and rejoining reaction referred to as telomere resolution ([4–6]; S1 Fig in S1 File). Telomere resolvases are a unique family of DNA cleavage and rejoining enzymes that promote telomere resolution at the *rTel* junctions. Four exemplars of the telomere resolvase family have been well characterized either by biochemical, genetic or structural means: TelA from *Agrobacterium tumefaciens*, ResT from *Borrelia burgdorferi*, TelK from the *Klebsiella* phage φKO2 and TelN from the *E. coli* phage N15 [7–10]. The telomere resolvases promote a reaction with mechanistic similarities to those promoted by type IB topoisomerases and tyrosine recombinases, utilizing a catalytic domain with structural similarity to the catalytic domains of these enzymes [10].

TelA, from *Agrobacterium tumefaciens*, has been characterized both biochemically and structurally [9, 11, 12]. Aside from promoting telomere resolution, TelA has also been reported to promote single-stranded DNA annealing reactions with both naked ssDNA and ssDNA complexed with its cognate single-stranded DNA binding protein [13]. These findings mirror those found for the *B. burgdorferi* telomere resolvase, ResT [14, 15]. The N-terminal domain of TelA is required for this activity but appears to be dispensable for telomere resolution [11, 16]. A TelA mutant with an N-terminal domain truncation can still promote telomere resolution but has lost much of the responsiveness to the presence of divalent metal ions ($Mg^{2+}$ or $Ca^{2+}$) normally observed with wild type TelA. The stimulatory effect of divalent metal ions appears to relieve autoinhibition by the N-terminal domain by affecting a step after substrate DNA binding [16]. This latter study suggested that the N-terminal domain also acts to restrain competing reactions that telomere resolvases have been shown to be capable of promoting: the reverse reaction of hp telomere fusion and the side reaction of generating Holliday junctions (HJ) via strand exchange between *rTels* [16–18]. A similar autoinhibitory role for the ResT N-terminal domain has been demonstrated that operates to mask substrate binding; rendering telomere resolution responsive to positive DNA supercoiling [19, 20].

In this study we used an AlphaFold2 generated model of a TelA monomer unbound to DNA to search for potential autoinhibitory interfaces in TelA responsible for regulating the activity of TelA. This modeling exercise identified two potential interfaces: one between the N-terminal domain and the catalytic domain encompassing residues 5–9 and residues 333–337; a second interface was potentially identified between the C-terminal helix, residues 421–442 and the hairpin-binding module helix 198–216 with charged residues R440/K441 potentially interacting with the D202 residue at the top of the hairpin binding module of TelA. By the use of targeted deletions and point mutations designed to weaken or abolish these interfaces we tested the model. We find that the model was mostly incorrect in its detailed predictions but was, nonetheless, extremely useful in guiding selection of mutations, that when combined, produced a telomere resolvase that was more than 250-fold activated for telomere resolution and that was largely independent of divalent metal ions.

## Materials and methods

### DNAs

All synthetic DNAs used to assemble substrates or produce site-directed mutations were purchased from Integrated DNA Technologies (IDT) and are listed in S1 Table in S1 File. All telomere resolution assays to derive initial rates used SspI-linearized pEKK392, the construction of which is reported elsewhere [16]. Telomere resolution reactions with a mutant *rTel* that has a CCATGA sequence between the scissile phosphates that inhibits hp formation, instead of a TCATGA wild type sequence, were conducted with pEKK495. pEKK495's assembly is reported elsewhere [12].

## Proteins

All TelA mutants reported in this study were generated by site-directed mutagenesis using oligonucleotides listed in S1 Table in S1 File. The induction and expression conditions are listed in S2 Table in S1 File. All TelA purifications proceeded as reported elsewhere, except with induction and expression conditions listed in S2 Table in S1 File [13, 16].

## Telomere resolution assays

Telomere resolution assays to derive initial rates of reaction were incubated at 0˚C, unless otherwise stated in the Figures and their legends. This reaction temperature ensured that we were able to measure rates for highly activated mutants. The reaction buffer contained 25 mM HEPES (pH 7.6), 1 mM DTT, 100 μg/mL BSA, 50 mM potassium glutamate, 1.8 μg/mL SspI linearized pEKK392 substrate DNA, 76 nM TelA and either 1 mM EDTA or 2 mM CaCl₂. A 120 μL master reaction was used. 18 μL aliquots were withdrawn at the indicated timepoints and the reaction was terminated by resuspension of the aliquots into SDS-containing load dye to a 1X concentration. 1X load dye contains 0.2% SDS, 20 mM EDTA, 3.2% glycerol, and 0.024% bromophenol blue. Subsequently, the timepoint samples were loaded to 0.8% agarose 1X TAE gels that were electophoresed at 3V/cm for 3 hours. The results were visualized by staining the gels with 0.5 μg/mL ethidium bromide for 30 min, followed by destaining in distilled water for 30 min. TelA (107–442), denoted as ΔN in the Figure legends, was found to be inactive at 0˚C, so the initial rate for this protein was determined in reactions performed as noted above, excepting that a 30˚C incubation temperature was utilized. Where addition of the N-terminal domain was added in *trans*, TelA (1–106), denoted as N, was added at a concentration of 800 nM. In order to assess the ability of TelA mutants to process a mutant *rTel* with a sequence (CCATGA as opposed to TCATGA found in wild type) between the scissile phosphates that inhibits hairpin formation we utilized reactions incubated at 30˚C utilizing pEKK495. The ability of hyperactive TelA mutants to resolve negatively supercoiled substrate plasmid, normally a very poorly reactive substrate, was assessed by reactions incubated at 30˚C for 60 min using 50 nM TelA and 2 μg/ml pEKK392. Gel images were visualized using a BioRad GelDoc system and the results were quantified with BioRad's QuantityOne software. Graphs were generated with Prism's GraphPad 6.0 software.

## Half-site cleavage and hairpin fusion assays

Half-site cleavage and hairpin formation assays were performed in a buffer containing 25 mM HEPES (pH 7.6), 1 mM DTT, 2 mM CaCl₂, 100 μg/mL BSA and 50 mM potassium glutamate. 76 nM TelA was incubated with 5 nM of 5' fluorescein-endlabeled parental half-site (OKBA27F/OGCB871) or the hp telomere (OKBA28F) at 30˚C for the times indicated (F indicates the labeled strand). Mock versions of the half-site and hp that maintain the sequence composition of the *bona fide* substrates were tested as well using oligonucleotides (OKBA29F/OGCB913; mock *rTel* and OKBA30F; mock hp). A 60 μL reaction volume was used and 18 μL aliquots were removed at the indicated timepoints and resuspended into a 1X concentration of an SDS-containing load dye. The samples were loaded to 8% PAGE, 1X Tris-acetate, EDTA (TAE), 0.1% SDS gels and were electrophoresed on 20 x 20 cm gels at 20V/cm for 150 min. These gels are native for DNA but denaturing for protein. The SDS allows the TelA covalently linked to the cleaved DNA to enter the gel. Gel images were visualized using a BioRad GelDoc system and the results were quantified with BioRad's QuantityOne software.

## Site-specific recombination assays

The ability of TelA mutants to promote recombination between *rTel*s was assayed using negatively supercoiled pEKK392 and a 5'-fluorescein end-labeled *rTel*. 190 nM of the indicated mutants were reacted with 10 μg/mL of pEKK392 and 70 nM oligonucleotide *rTel* (OGCB951F/OGCB952) in our standard telomere resolution reaction buffer with 2 mM $CaCl_2$ (see above). The oligonucleotide *rTel* was partitioned between 20 nM of fluorescein labeled oligonucleotide *rTel* and 50 nM of unlabeled *rTel*. Control incubations omitted TelA. Reactions were incubated at 30˚C for 2 h followed by the addition of 5X SDS loading dye to a 1X final concentration prior to gel loading. Reactions were loaded to 0.8% (w/v) agarose 1X TAE gels and electrophoresed at 1V/cm for 19 h. The bottom of the gel with the free unreacted *rTel*s were cut off and then gels were imaged for fluorescein labeled material. Subsequently, gels were stained with 0.5 μg/mL ethidium bromide and visualized on a Biorad GelDoc system using the ethidium bromide program to visualize the plasmid substrate and molecular weight markers.

# Results

## Modeling of potential autoinhibitory interfaces within TelA

There has been a co-crystal structure of TelA with various substrates and reaction products available for several years [9]. While extremely informative the structures, unfortunately, do not resolve the N-terminal domain or show a structure of the enzyme unbound to DNA. Since the remarkable improvement in structure modeling reliability represented by AlphaFold2 we were interested in modeling TelA as an unbound monomer to see if we could discover the molecular basis for the N-terminal domain's autoinhibitory role on telomere resolution [16, 21]. The top ranked model generated by AlphaFold2 is shown in Fig 1A alongside a monomer view of TelA bound to a hairpin (hp) telomere. AlphaFold2 predicts the structure of the N-terminal domain (shown in red) and shows it to stack upon the catalytic domain of TelA (pink) in a manner that, potentially, could interfere with activity. Additionally, the C-terminal helix (shown in green) is predicted to switch from a conformation found in TelA bound to substrate or products in which the C-terminal helix is swapped between protomers of the active dimer to a conformation that interacts with the hairpin binding module helix of the TelA N-core domain in the monomer. This helix in the N-core domain has many critical residues needed for telomere resolution and could, therefore, represent another autoinhibitory interface [12, 22, 23]. Examination of the details of the interfaces in the AlphaFold2 model reveals the juxtaposition of oppositely charged residues across the interfaces (Fig 1B). This suggested that there may be two autoinhibitory interfaces in the TelA monomer maintained, in part, by electrostatic interactions across the interfaces. We focused on the potential electrostatic interactions, as this would allow us to employ the method of neutralizing and reversing charges and to map potential interacting pairs via charge reversals in double mutant contexts. The predicted aligned error plot for the model is presented in S2 Fig in S1 File and indicates a low confidence (~30 Å) in the prediction for interface 1 but much higher confidence for interface 2 (~10 Å).

## Mutation of interfaces 1 & 2 stimulates telomere resolution

In order to investigate whether the model was predicting real autoinhibitory interfaces, we generated, expressed and purified TelA mutants designed to weaken or interrupt the hypothesized interfaces. To assess the effects of these mutations on TelA activity we measured the initial rate of reaction of each mutant under conditions without divalent metal ion (1mM EDTA) or in the presence of 2 mM $Ca^{2+}$. We measured the rates at 0˚C instead of the standard 30˚C

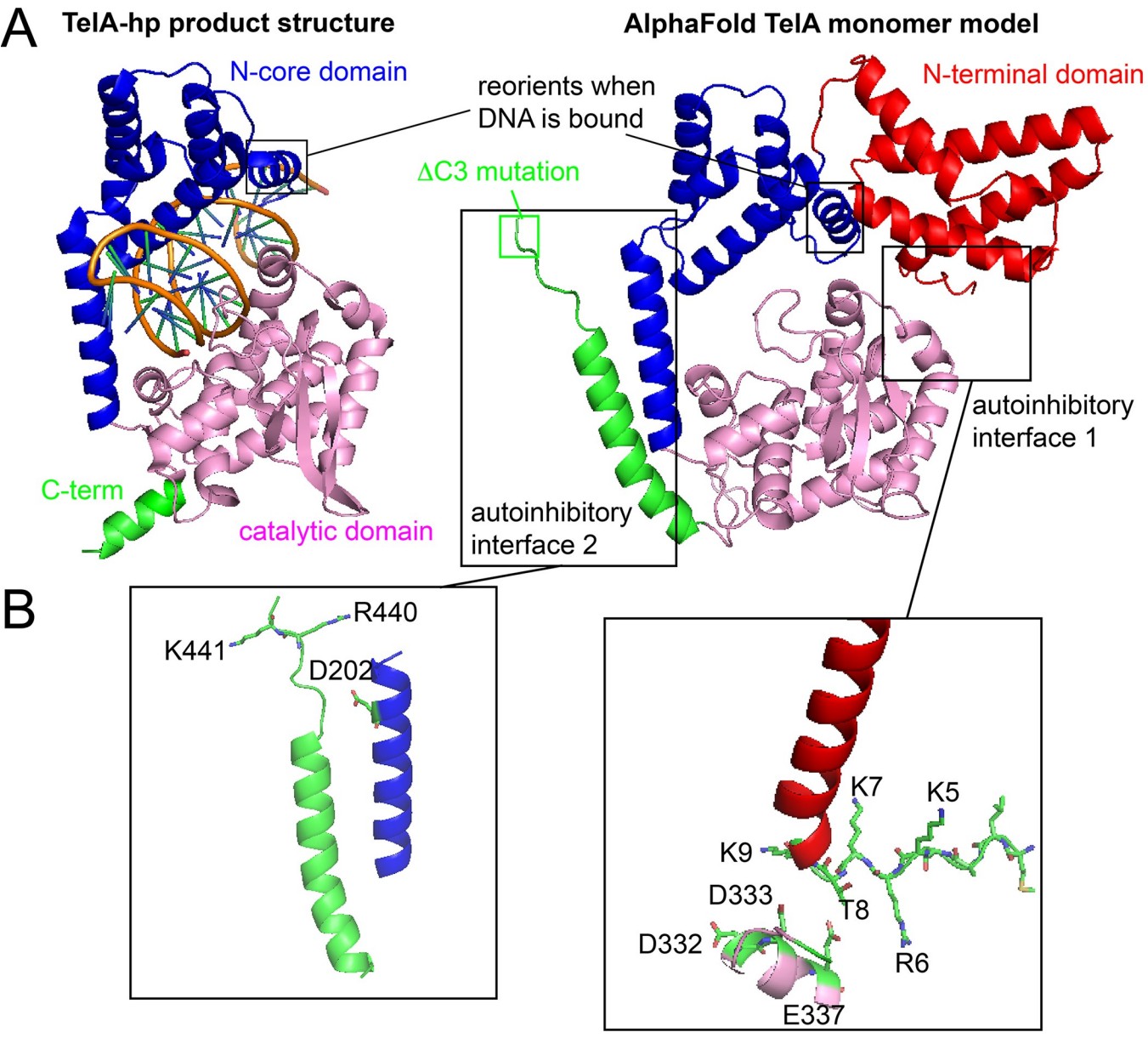

**Fig 1. Structural modeling of potential TelA autoinhibitory interactions.** A) TelA-hairpin (hp) product complex structure (one protomer of the dimer shown) *vs.* an AlphaFold2 model of a TelA monomer without bound DNA. In the model the positively charged beginning of the N-terminal domain interacts with an acidic patch in the catalytic domain (interface 1) and the C-terminal helix reorients to form the autoinhibitory interface 2. The N-terminal domain (red) is encompassed by residues 1–106, the N-core domain (blue) by residues 107–217, the catalytic domain (pink) by residues 218–410 and the C-terminal helix (green) by residues 411–442. Within the N-core domain the hairpin binding module lies between residues 197–216 and comprised the long alpha helix that connects the N-core to the catalytic domain. The model derived from the structural data was generated using PyMol and PDB accession # 4e0g. The model derived from AlphaFold2 was generated using using PyMol and AlphaFold2 [21]. B) Details of the autoinhibitory interfaces with charged residues of interest shown.

reaction temperature in order to slow reactions down sufficiently so that we could determine initial rates for hyperactive mutants.

We employed charge neutralization (to alanine), charge reversal or deletion of the charged residues at each interface (Fig 2). At interface 1 the N-terminal domain residues K5, R6 and K7 were examined by charge reversal mutations to D or E residues. This afforded modest stimulation (1.4–2.7-fold) of the reaction under conditions with $Ca^{2+}$ present but were not

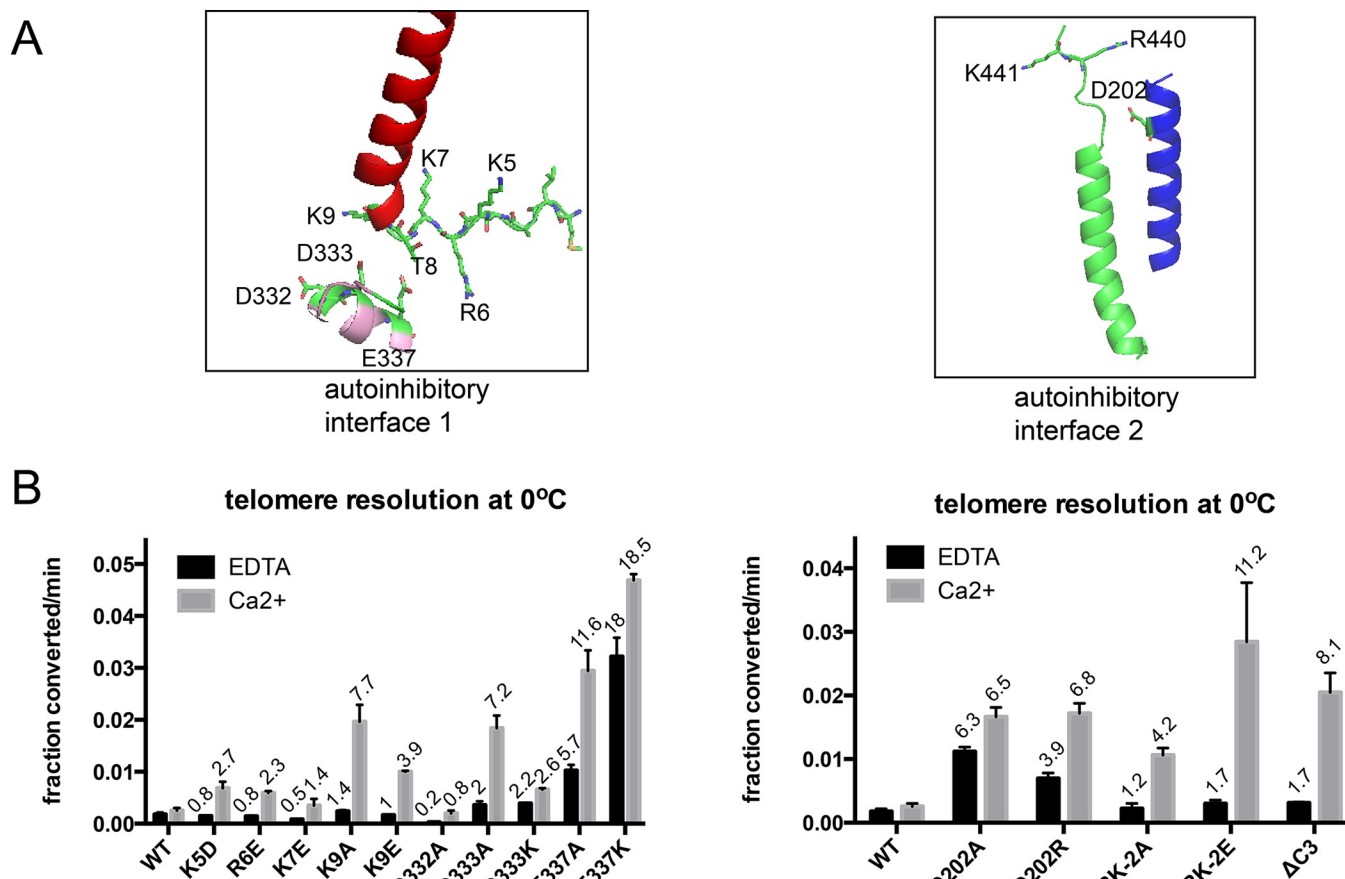

**Fig 2. Telomere resolution assays of interface 1 & 2 mutants.** A) Details of the potential autoinhibitory interfaces revealed in an AlphaFold2 model of the TelA monomer. B) Graphical summary of the initial rates of telomere resolution conducted at 0°C showing the rates of reactions incubated without divalent metal ions (1 mM EDTA) compared to the rates of reaction in the presence of 2 mM CaCl$_2$. Shown is the mean and standard deviation of 3 independent replicates. The values over the individual bars represent the fold stimulation over the equivalent conditions for wild type TelA.

stimulatory under conditions with EDTA (Fig 2A). We took these results to indicate that the mutations had weakened the interface enough to be apparent under metal-stimulation but not sufficiently to activate TelA in the absence of divalent metal ions. Amongst residues of interest in the N-terminal domain, mutation of the K9 residue to A or E led to the greatest stimulation of the reaction compared to wild type TelA, affording 7.7-fold and 3.9-fold stimulation, respectively, in conditions containing Ca$^{2+}$. On the other side of interface 1 the D332, D333 and E337 residues were mutated to neutralize and/or reverse the charge of the residue. The D332A mutation was less active than wild type at 0°C and was, therefore, inferred not to contribute to the interface. D333A provided stimulation similar to that afforded by the K9A mutation and D333K still provided modest stimulation in conditions with Ca$^{2+}$ (7.7 and 2.6-fold). The greatest stimulatory effect of single point mutations at interface 1 was found with mutation of E337 to A or K, affording 11.6-fold and 18.5-fold stimulation of the reaction rate with Ca$^{2+}$, respectively. Interestingly, the calcium dependence of the reaction seems attenuated with the E337K mutant with a robust 18-fold stimulation of the reaction observed under EDTA conditions.

At interface 2, charge neutralization and charge reversal of residue D202 of the N-core domain both afforded ~6-fold stimulation of the reaction and, as expected from previous results, reduced the calcium dependence of telomere resolution (Fig 2B; see also [16]). The C-terminal helix terminates in RKG residues that may be in place for interactions with D202. We

generated a deletion mutant that truncated TelA to delete these last 3 residues (ΔC3) and found this mutant to provide 8.1-fold reaction stimulation relative to wild type TelA, with $Ca^{2+}$. Charge neutralization or charge reversal of both the R440 and K441 residues in tandem produced 4.2-fold and 11.2-fold stimulation with $Ca^{2+}$, respectively, in the resulting double mutants.

Because the mutations we generated at the hypothesized autoinhibitory interfaces were found to be mostly stimulatory there was less concern that the mutations had led to their phenotypes by inducing misfolding of the resulting proteins. Nonetheless, we assessed the effect of the mutations on other activities of TelA to control for such potential effects. TelA has been characterized to be a robust single-stranded DNA annealing protein and it is known that a properly folded, full-length TelA is required for optimal annealing [13]. Therefore, we assayed our mutants with a stringent annealing assay employing the structured HIV transactivational response element (HIV$_{TAR}$; S3 and S4 Figs in S1 File). Single mutants of the two interfaces showed no dramatic hypo- or hyperactivity in the annealing assays.

## Combining activating mutations leads to additive activation of TelA

The characterization of TelA mutants that weakened the hypothesized interactions on one side of the two interfaces produced a range of activation from 2-18-fold. Therefore, it seemed plausible that both interfaces contributed to autoinhibition of TelA's telomere resolution activity. If the two interfaces are independent of each other one would predict that the effect of combining activating mutations from the two interfaces would be additive (*i.e.* greater than the activation afforded by either mutation alone). If, instead, the mutations affected a common interface, combining mutations would not be expected to result in activation greater than the most activating single mutation. To test this prediction we generated double mutants that combined activating mutations from interface 1 and interface 2 (Fig 3).

We first tested the effect of combining the ΔC3 and D202A or D202R mutations. Since the interface 2 model posits these residues to be interacting to help form the interface we were surprised to find that the activation afforded by the ΔC3 and D202A or D202R mutations to be mildly additive. This may reflect the role of D202 in reaction responsiveness to divalent metal ions [16]. Therefore, in an effort to obtain optimally activated mutants both (ΔC3; D202A) and (ΔC3; D202R) were combined with the optimally activating interface 1 mutation E337K.

The combination of the interface 1 (E337K) mutation with either of the interface 2 mutations (ΔC3; D202A and ΔC3; D202R) resulted in TelA variants that exhibited additive stimulation (55-124-fold stimulation). The (ΔC3; D202RE337K) mutant, in particular, shows greater than 89-fold stimulation of the initial rate in EDTA conditions and a 124-fold stimulation, over wild type TelA, in $Ca^{2+}$ buffer conditions. The combination of activating mutations from the two hypothesized interfaces with mutations that, simultaneously, reduce the divalent metal ion dependence of telomere resolution has produced a largely metal insensitive and hyperactive mutant (Fig 3B). As expected, the tested mutants were found to have approximately wild type levels of annealing proficiency indicating no gross misfolding issues (S4 Fig in S1 File).

## Testing the modeled autoinhibitory interface interactions with paired charge reversal mutagenesis

If the details of the modeled autoinhibitory interfaces in TelA are accurate, one would predict that combining charge reversal mutations on both sides of the interfaces would result in suppression of the activation observed with single charge reversal mutants by reinstating electrostatic interactions at the proposed interfaces. To test the possible charge-charge interactions at interface 1 we combined charge reversal mutations R6E, K7E and K9E with either D333K or

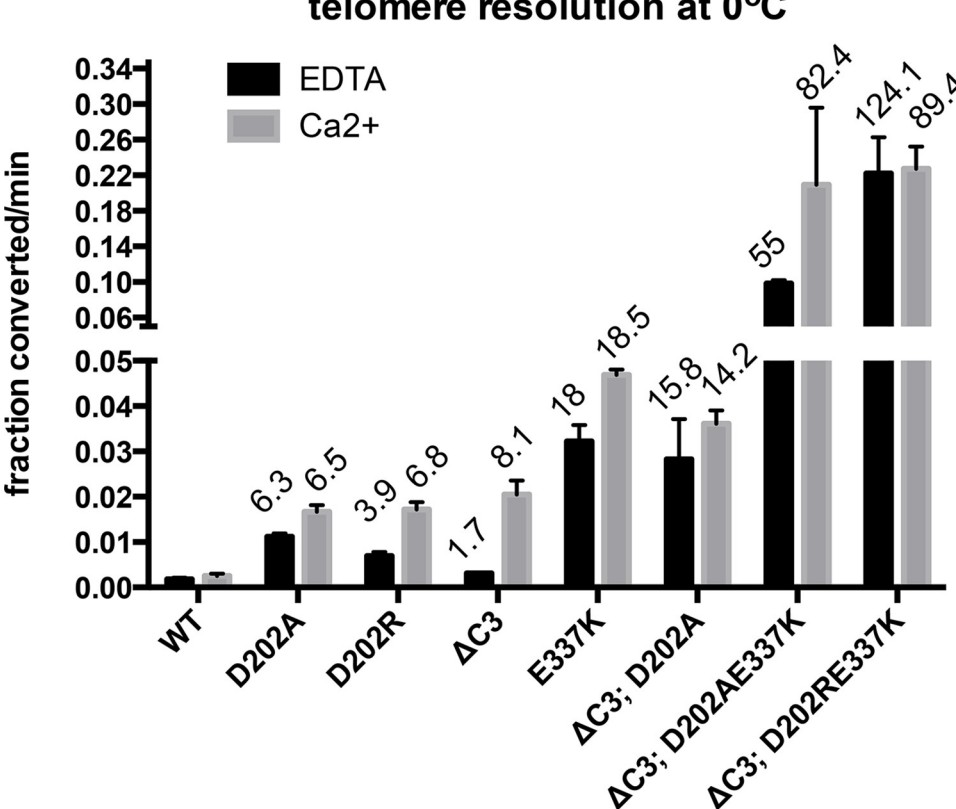

**Fig 3. Telomere resolution assays of interface 1 & 2 double mutants.** Graphical summary of the initial rates of telomere resolution conducted at 0°C with 1 mM EDTA *vs.* 2 mM CaCl₂. Shown is the mean and standard deviation of 3 independent replicates excepting the ΔC3; D202AE337K and ΔC3; D202RE337K mutants that were assayed 6 times. ΔC3 denotes the deletion of the last 3 amino acids residues from the C-terminal domain. This removes R440 and K441 as well as the terminal glycine of TelA. The fold stimulation over wild type is shown.

E337K (Fig 4). In both cases there was a pattern of activity where R6E had the least effect on activity, K7E had an intermediate effect and K9E reduced activity the most when combined with either D333K or E337K. This suggested that K7E and K9E interacted with D333K and E337K. However, examination of the double mutants in the annealing assay revealed hypoactivity in this assay for both the (K9ED333K) and (K9EE337K) mutants, indicating that these proteins are likely improperly folded (S4 Fig in S1 File). The (K9E), (D333K) and (E337K) single mutants were wild type in the annealing assay (S3 Fig in S1 File). In aggregate, these results provide some evidence that the K7 residue may interact with D333 and/or E337 in TelA to establish autoinhibitory interface 1.

To test possible interactions at interface 2 we combined the D202R mutation with the dual charge reversal mutations of the terminal RK motif at the end of TelA (R440EK441E). The (R440EK441E) mutant showed a robust 11.2-fold activation over wild type TelA. The combination of the D202R mutation with R440EK441E, unexpectedly, resulted in further activation of TelA (88.4-fold with EDTA and 61.6-fold activation with Ca²⁺ reaction conditions; Fig 4). This combination of mutations has eliminated the divalent metal-dependence of the reaction. The (D202RR440EK441E) triple mutant was found to have wild type proficiency in annealing assays (S5 Fig in S1 File). These results do not support the hypothesis that D202 interacts with R440 and/or K441 to establish autoinhibitory interface 2.

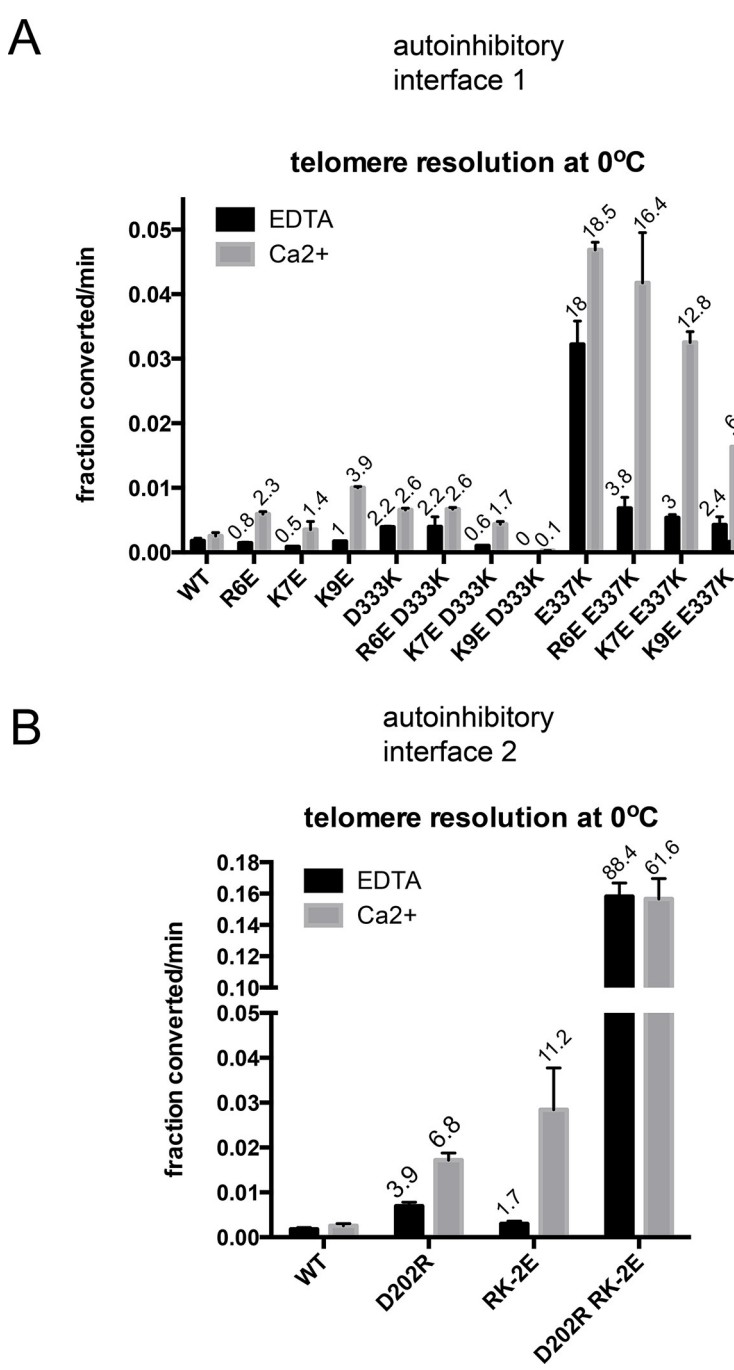

**Fig 4. Telomere resolution assays of paired charge reversal mutants.** A) Graphical summary of the initial rates of telomere resolution conducted at 0˚C with 1 mM EDTA *vs.* 2 mM CaCl$_2$ of interface 1 charge reversal single and double mutants. Shown is the mean and standard deviation of at least 3 independent replicates. B) Graphical summary of the initial rates of telomere resolution conducted at 0˚C with 1 mM EDTA *vs.* 2 mM CaCl$_2$ of interface 2 charge reversal mutants. Shown is the mean and standard deviation of 3 independent replicates, excepting for the R6EE337K and K9EE337K mutants that had 6 replicates. RK-2E denotes the TelA double charge reversal mutations R440EK441E. The fold stimulation over wild type is shown.

### Testing the modeled autoinhibitory interface 1 interactions with deletion analysis and N-terminal domain add back experiments

The use of paired charge reversal mutations across the putative autoinhibitory interfaces provided modest evidence of potentially interacting residues at interface 1. In order to provide a more stringent test of the proposed interactions we assessed the effect of the most stimulatory interface 1 point mutation (E337K) in a background where the interacting N-terminal domain has been deleted (ΔN; E337K). Measuring the initial rate at 0˚C revealed that the ΔN mutant was unreactive at this reaction temperature. However, the double mutant (ΔN; E337K) showed the greatest degree of reaction activation seen yet with any of our mutants (>200-fold in both EDTA and $Ca^{2+}$ conditions; Fig 5B). This strongly suggested that the E337K mutation must activate the reaction, at least in part, by mechanisms independent of a contribution to the hypothesized autoinhibitory interface 1.

To reconfirm our previous study that adding the N-terminal domain, in *trans*, to reactions with the ΔN mutant leads to reaction inhibition we measured the initial rate of reaction of the (ΔN; E337K) mutant with a 10-fold molar excess of the N-terminal domain added, in *trans*

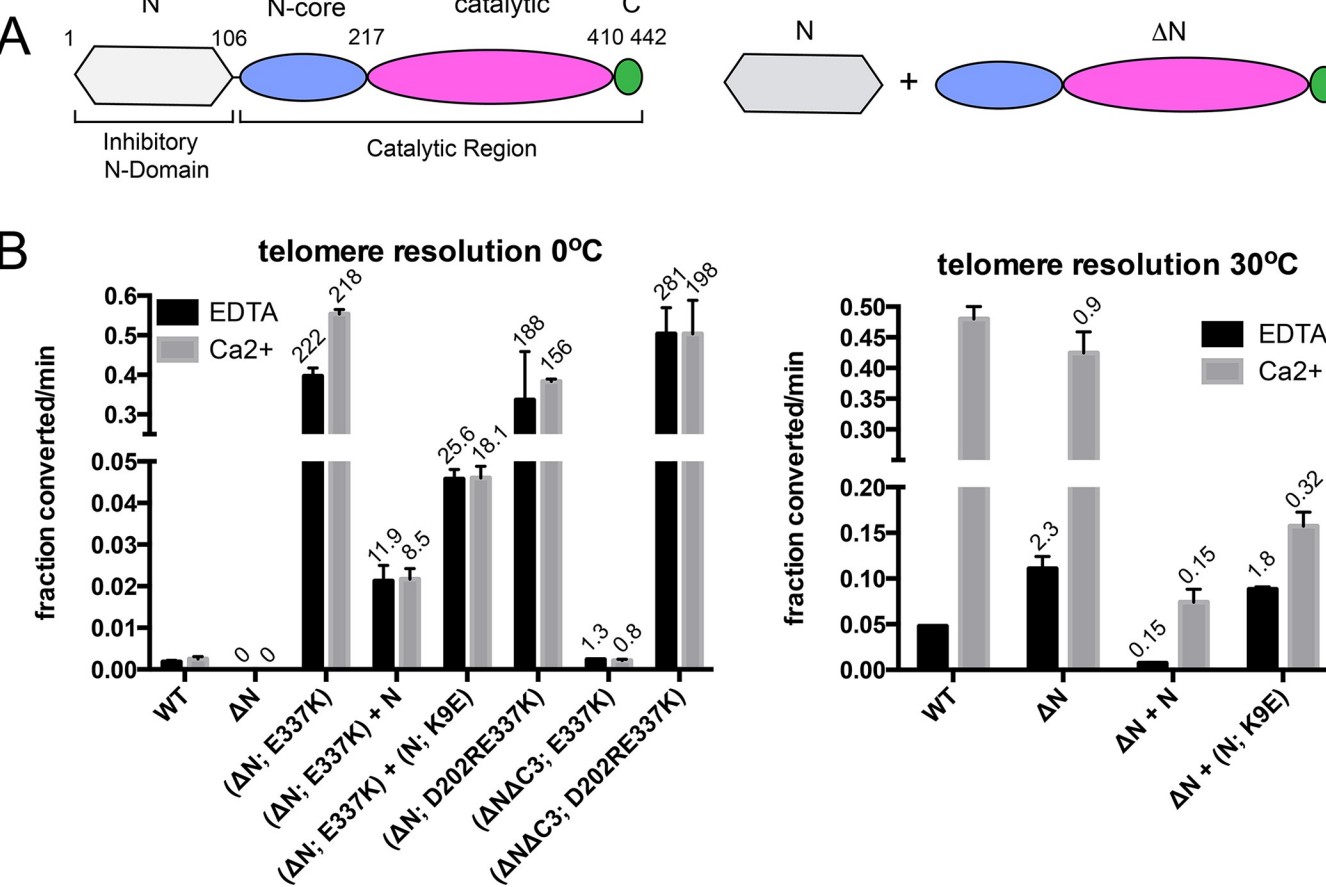

**Fig 5. Telomere resolution by TelA N-terminal domain deletion mutant combined with the interface 1 mutant E337K.** A) Schematic summary of the domain structure of TelA followed by a schematic of the approach of combining the N-terminal domain deletion mutant of TelA (ΔN) with an excess of the N-terminal domain (N) added in *trans* to assess the inhibitory potential of the exogenously added N-terminal domain. Residue numbers delimiting the TelA domains is shown in the first graphic. B) Graphical summary of the initial rates of telomere resolution conducted at 0˚C (left graph) or 30˚C (right graph) using buffers containing 1 mM EDTA *vs.* 2 mM $CaCl_2$. Shown is the mean and standard deviation of 3 independent replicates excepting the ΔN; D202RE337K mutant that had 6 replicates. Where the N-terminal domain is added in *trans* (+N) it is added in 10.5-fold molar excess over the ΔN mutant. The fold stimulation over wild type is shown.

([16]; Fig 5B). Adding the N-terminal domain (N) reduced the reaction rate of (ΔN; E337K) ~22-fold, indicating that the N-terminal domain does have the ability to inhibit TelA activity. Testing the effect of adding the N-terminal domain harbouring the activating mutation K9E, in *trans*, (N; K9E) resulted in a lesser degree of reaction inhibition (~10-fold) compared to the ~22-fold reduction afforded by adding wild type N. We assessed the effect of adding (N) and (N; K9E) to reactions with (ΔN) by running assays at the standard reaction temperature of 30˚C (Fig 5B). The same pattern of inhibition seen with the (ΔN; E337K) mutant was obtained with (ΔN), except that there was a greater dependence on inhibition of having $Ca^{2+}$-free conditions employed, consistent with the effect of the E337K mutation lowering the $Ca^{2+}$-dependence of telomere resolution (Fig 2).

The contradictory results from the activation afforded by combining ΔN with E337K to make a double mutant (activation) *vs*. combining ΔN and E337K in N-terminal domain add back experiments (inhibition) highlighted the possibility that the phenotype caused by the E337K and K9E mutations could be due to factors distinct from weakening the hypothesized interface 1. We examined (ΔN) and (ΔN; E337K) in annealing assays and confirmed previous results that the presence of the N-terminal domain is necessary for annealing reactions (S5B Fig in S1 File; [16]). Using the bulged hairpin structured $HIV_{TAR}$ element we also found the N-terminal domain on its own to be a poor annealing protein when provided with substrates with strong secondary structure. The N-terminal domain with the K9E mutation (N; K9E) showed even worse annealing activity than N (S5B Fig in S1 File). We have never been able to detect bandshifts of substrate DNA by (ΔN) in electromobility shift assay (EMSA) experiments and this continued to be the case in this study. However, we did detect bandshifts using (ΔN; E337K). S6B, S6C Fig in S1 File documents EMSA experiments with (ΔN) and (ΔN; E337K) using substrate replicated telomeres (*rTels*) and hp products. We infer from the pattern of binding activity that (ΔN; E337K) resolves the *rTel* and that it is the hp product that (ΔN; E337K) was binding to produce a bandshift in our assays. (ΔN) was unable to bandshift either *rTel* or hp product (S6B, S6C Fig in S1 File). We also examined the N-terminal domain (N) and (N; K9E) in EMSA experiments to determine if the K9E mutation had affected the non-specific DNA binding activity of the N-terminal domain (S6D Fig in S1 File). (N) shows a laddering bandshift indicative of successively more N loading onto the *rTel* substrate. However, (N; K9E) failed to bind DNA; this is consistent with the abrogated annealing activity seen in S5B Fig in S1 File. We had established in a previous study that interaction of (N) with (ΔN) in crosslinking studies was promoted by the presence of DNA [16]. It seems likely that the lesser degree of reaction inhibition seen with (N; K9E) added, in *trans*, to (ΔN) and (ΔN; E337K) is due to this loss of DNA binding activity caused by the K9E mutation. Finally, we assessed the behaviour of another of our hyperactive TelA mutants (ΔC3; D202RE337K) in an EMSA experiment with an *rTel vs*. wild type TelA (S6D Fig in S1 File). Wild type TelA produced c1 and c2 complexes inferred to be the binding of one protomer (c1) and two protomers of TelA (c2) while the hyperactive (ΔC3; D202RE337K) mutant again resolved the *rTel* and bandshifted the hp product (c3). This indicates that the binding behavior of the (ΔN; E337K) is a general feature of activation by the E337K mutation rather than a feature specific to the ΔN background.

Finally, we expected that combining our most activating mutations together would produce optimally activated TelA variants. We tested this by examining (ΔN; D202RE337K), (ΔNΔC3; E337K) and (ΔNΔC3; D202RE337K) mutants for reaction stimulation (Fig 5B). Both (ΔN; D202RE337K) and (ΔNΔC3; D202RE337K) were highly activated for both EDTA and $Ca^{2+}$ conditions but (ΔNΔC3; E337K) was poorly active. This unexpectedly poor activity is likely due to a folding issue as it is the only protein of the three tested that had defects in annealing and substrate binding (S7 Fig in S1 File).

## Does hyperactivation of TelA unleash competing or inappropriate activities?

We assessed the effect of hyperactivation of TelA on competing activities with a collection of mutants that showed >50-fold activation of the initial rates of reaction. First we wanted to test these mutants on a negatively supercoiled substrate plasmid since it is known to be a poor substrate for resolution; resolution normally requires either relaxation of the supercoils or linearization of the substrate DNA (Fig 6A; [6, 11]). Wild type TelA and most of the mutants showed little activity in this assay. The highly activated (ΔN; D202RE337K) and (ΔNΔC3; D202RE337K) mutants were, unexpectedly, able to resolve the supercoiled substrate. The gel shows a ladder of topoisomers indicating that these mutants may have relaxed the plasmid enough to be able to resolve it.

We next tested if the hyperactive mutants tried to resolve a mutant *rTel* that has been asymmetrized between the scissile phosphates; thereby inhibiting hairpin formation. None of the hyperactivated mutants showed inappropriate activity on this substrate (that would produce TelA blocked double-strand breaks). Reaction with a negatively supercoiled version of this plasmid substrate revealed topoisomerase activity for each mutant on this substrate; indicating that they are all active but locked into abortive DNA cleavage and rejoining reactions (S8 Fig in S1 File).

We also tested if the competing side reaction of site-specific recombination between *rTel*s was unmasked by the activating mutations (Fig 6C). Wild type TelA is prohibited from promoting this reaction, in part, by autoinhibition by the N-terminal domain [16]. The (ΔC3; D202AE337K) and (ΔN; E337K) mutants were also inactive for recombination but the remaining mutants were all able to form Holliday junctions. This pattern of reactivity suggests that the D202 residue may also play a role in masking this latent activity and that reversal of the charge at this position can also activate this activity (Fig 6C). It is unclear why combining the E337K mutation with the N-terminal domain deletion has suppressed the recombination that we know the ΔN mutation allows [16].

Finally, we assessed whether the activating mutations in the hyperactive TelAs had unleashed the latent ability of the enzyme to promote the reverse reaction of hp telomere fusion [16, 17]. Fig 7A shows an analysis with telomeric suicide half-sites to assess the DNA cleavage activity of wild type TelA and the hyperactive TelA mutants. The incubation was at 5˚C overnight, a condition that favours the reverse reaction. All proteins showed a relatively uniform level of DNA cleavage activity with the telomeric half-site (32–40%). In the assay with the hp telomere our positive control for telomere fusion was provided by the reaction of the (ΔN) mutant supplemented with excess N-domain provided in *trans* (59% conversion; [16]). Most of our hyperactivated mutants were also activated for reaction reversal without the need for adding excess N-terminal domain (18.4–38.5% conversion). Wild type TelA and the (ΔC3; D202AE337K) and the (ΔN; E337K) mutants were the exception; being hyperactivated for telomere resolution without unmasking the latent telomere fusion reaction (Fig 7B). All TelA variants were inactive when tested with mock telomeric half-site and hairpin substrates, confirming the site-specificity of all the hyperactivated versions of TelA (Fig 7A & 7B).

## Testing the most useful hyperactive mutants at standard reaction temperature

Our analysis from Figs 6 & 7 indicated that the (ΔC3; D202AE337K) and (ΔN; E337K) mutants were successfully hyperactivated without unleashing inappropriate competing reactions. The telomere resolution activity of all our mutants was assessed at the unusual reaction temperature of 0˚C to facilitate easy comparison of the initial rates for proteins spanning a wide range

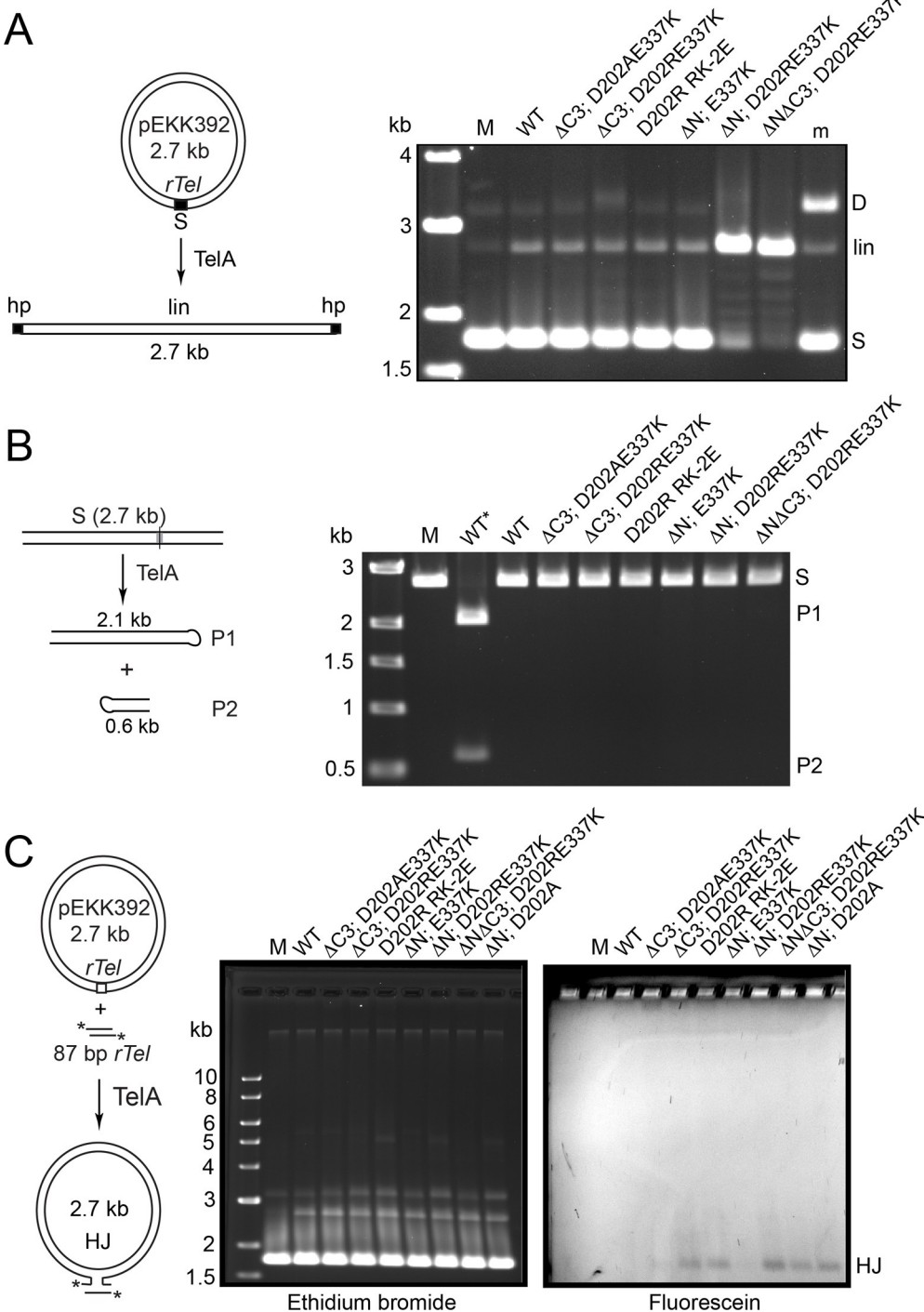

**Fig 6. Assessing the ability of hyperactive TelA mutants to promote aberrant reactions.** A) Reaction of hyperactive TelA mutants with a negatively supercoiled *rTel* plasmid substrate (pEKK392). On the left is a graphic of telomere resolution (where it occurs) of the plasmid substrate. On the right is presented a gel panel of an 0.8% agarose 1X TAE gel of reactions incubated with 50 nM of the indicated proteins and 2 μg/ml plasmid substrate at 30°C for 60 min. M denotes a mock reaction without added TelA; WT denotes wild type TelA; m denotes a marker for supercoiled monomer substrate (S), linear substrate (lin) and supercoiled dimer (D). B) Reaction of hyperactive TelA mutants with a linearized mutant *rTel* that suppresses hp telomere formation by asymmetrization of the sequence between the scissile phosphates. On the left is a graphic of the telomere resolution reaction (where it occurs). On the right is a 0.8% agarose 1X TAE gel panel of reactions incubated with 76 nM of the indicated proteins and 2 μg/ml plasmid substrate at 30°C for 30 min. M denotes a mock reaction without added TelA; WT* denotes wild type TelA reacted with a wild

type *rTel* plasmid; WT denotes wild type TelA reacted with the mutant *rTel* plasmid (pEKK495); S denotes the linear substrate DNA; P1 & P2 denote the products of telomere resolution. C) Reaction of hyperactive TelA mutants with a negatively supercoiled *rTel* plasmid and a fluorescently labeled short synthetic *rTel*. On the left is a graphic detailing HJ formation between the synthetic and plasmid *rTel*s. On the right are 0.8% agarose 1X TAE gel panels of reactions of incubated with 190 nM TelA and 10 μg/ml plasmid substrate and 20 nM labeled synthetic *rTel* and 50 nM unlabeled *rTel* incubated at 30°C for 120 min.

of rates (Figs 2–5). We repeated the telomere resolution analysis of this pair of utile hyperactive mutants and compared their activation relative to wild type TelA in reactions conducted at the standard reaction temperature of 30°C. In order to slow down the reaction sufficiently to measure rates we reduced the TelA concentration in the assay from the 74 nM concentration used at 0°C to 3 nM for assays conducted at 30°C (Fig 8). Wild type TelA did not have detectable

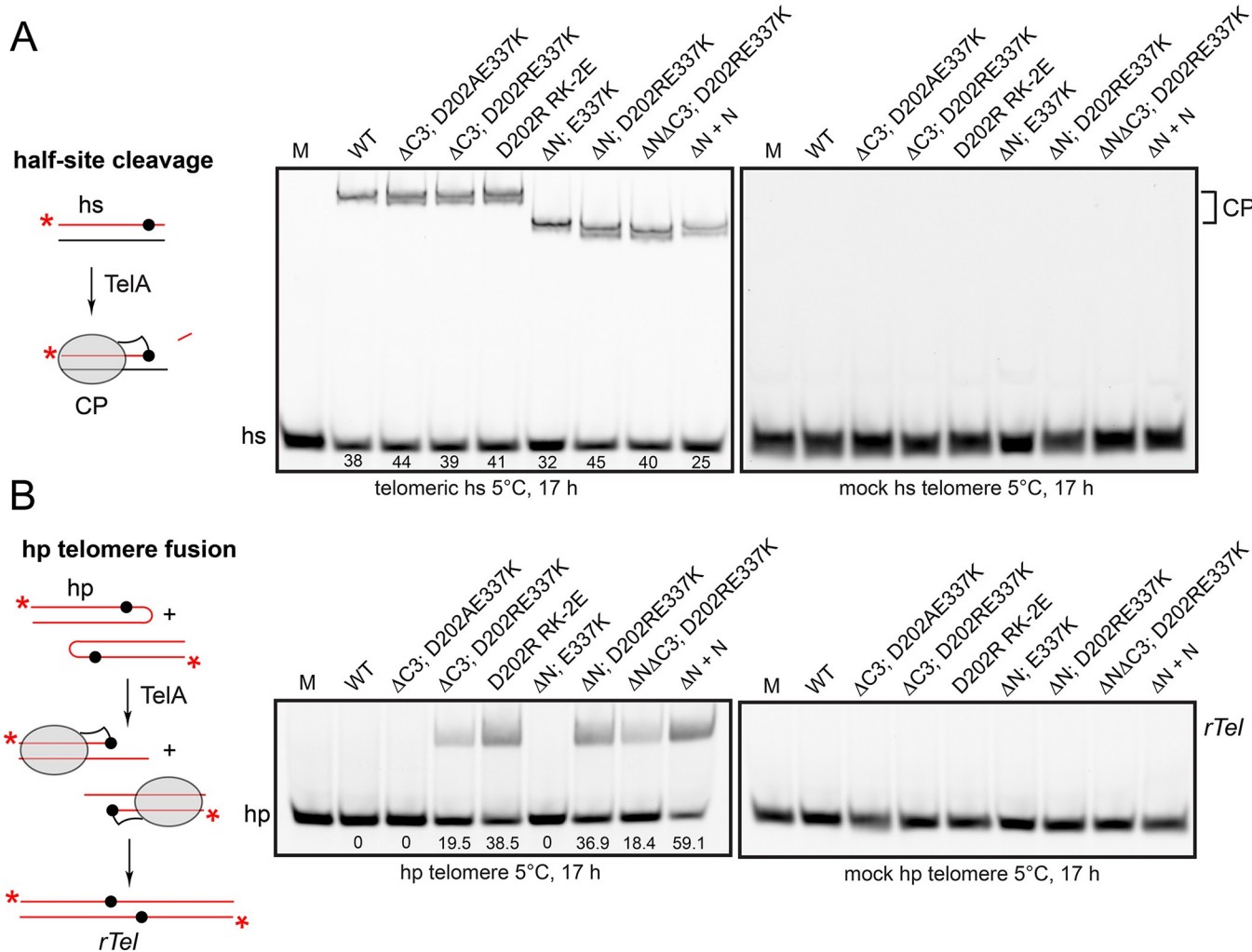

**Fig 7. Assessing the ability of hyperactive TelA mutants to promote reaction reversal.** A) Half-site cleavage assay with hyperactive TelA mutants. On the left is a graphic of the cleavage reaction in which TelA becomes covalently trapped on a suicide half-site after DNA cleavage. On the right are 8%PAGE 1X TAE/0.1% SDS gel panels run with 150 nM of the indicated hyperactive TelA mutants incubated at 5°C for 17 h. The numbers below the lanes in the telomeric half-site panel reflect percent DNA cleavage of the half-site. hs denotes halfsite; CP denotes cleavage products. The difference in mobility of the cleavage products is due to the smaller size of the ΔN mutants. B) Telomere fusion assays with hyperactive TelA mutants. On the left is a graphic of the hp telomere fusion reaction with a hp telomere producing an *rTel* junction (reaction reversal). On the right are 8%PAGE 1X TAE/0.1% SDS gel panels run with 150 nM of the indicated hyperactive TelA mutants at 5°C for 17 h. The numbers below the lanes in the hp telomere panel reflect percent conversion from hp telomeres to fused *rTel* junctions. hp denotes hairpin telomere; *rTel* denotes a replicated telomere junction formed by fusing hp telomeres; CP denotes cleavage products.

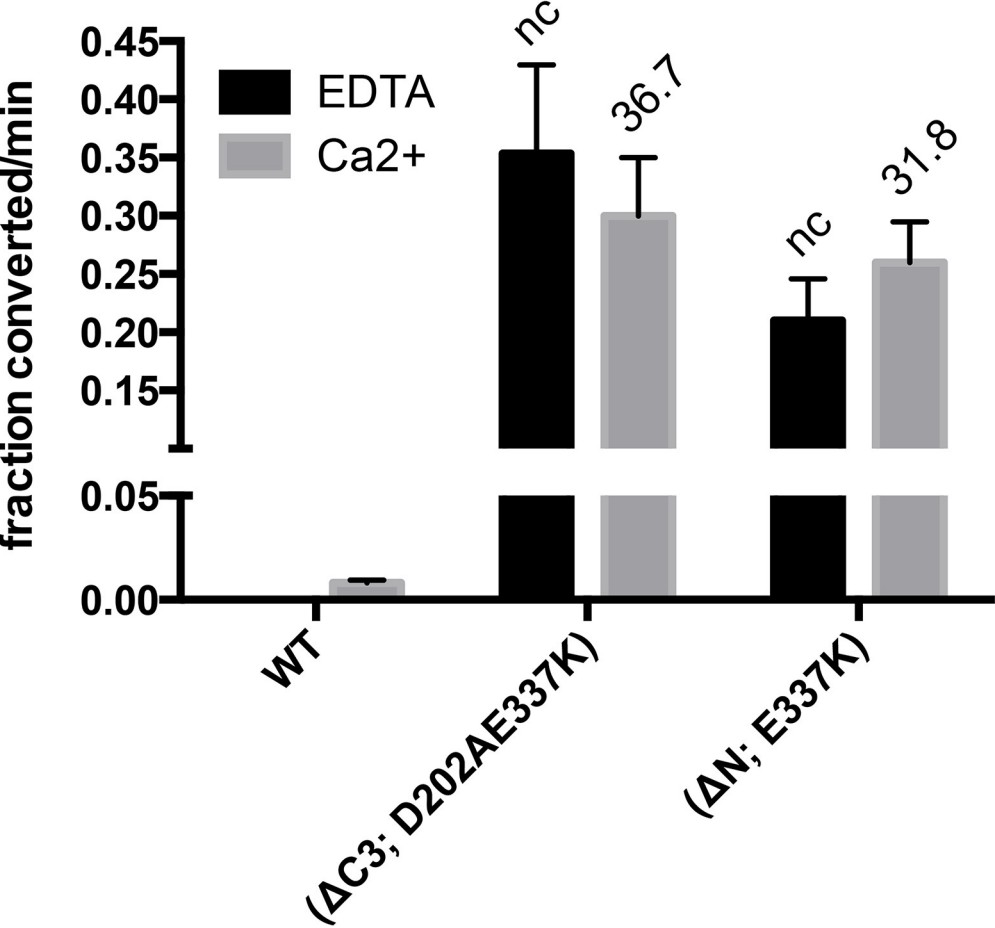

**Fig 8. Telomere resolution assays of hyperactive mutants at low protein concentration.** Graphical summary of the initial rates of telomere resolution conducted at 30˚C using 3 nM TelA showing the rates of the reaction incubated without divalent metal ions (1 mM EDTA) compared to the rates of reaction in the presence of 2 mM CaCl$_2$. Shown is the mean and standard deviation of 3 independent replicates. The values over the individual bars represent the fold stimulation over the equivalent conditions for wild type TelA. nc denotes not calculated since the fold stimulation for the EDTA conditions were not calculated due to the lack of measurable activity for wild type TelA in EDTA.

activity in EDTA conditions at this low TelA concentration but had a measurable rate with Ca$^{2+}$ containing conditions. Both hyperactive mutants showed robust hyperactivation at 30˚C indicating that the hyperactivity of these mutants was not, primarily, due to a relief of the cold-sensitivity of the telomere resolution reaction. It is noteworthy that differential sensitivity to cold reaction temperatures does play some role in the observed reaction stimulation as the (ΔC3; D202AE337K) mutant is more active than the (ΔN; E337K) mutant at 30˚C whereas the opposite was true at 0˚C.

A summary of the assay results for all tested mutants is presented in Table 1.

## Discussion

In the present study we have interrogated an AlphaFold2 model of a TelA monomer unbound to DNA that predicted the presence of two autoinhibitory interfaces. The predicted aligned

**Table 1. Summary of mutant phenotypes.**

| Protein | tel. res. [1] | hp fusion | recombin. | topo. | annealing | rTel bind |
|---|---|---|---|---|---|---|
| wild type | 1 | no | no | no | normal | normal |
| K5D | 2.7 | N/T | N/T | N/T | normal | N/T |
| R6E | 2.3 | N/T | N/T | N/T | normal | N/T |
| K7E | 1.4 | N/T | N/T | N/T | normal | N/T |
| K9A | 7.7 | N/T | N/T | N/T | normal | N/T |
| K9E | 3.9 | N/T | N/T | N/T | normal | N/T |
| D332A | 0.8 | N/T | N/T | N/T | normal | N/T |
| D333A | 7.2 | N/T | N/T | N/T | normal | N/T |
| D333K | 2.6 | N/T | N/T | N/T | normal | N/T |
| E337A | 11.6 | N/T | N/T | N/T | normal | N/T |
| E337K | 18.5 | N/T | N/T | N/T | normal | N/T |
| D202A | 6.5 | N/T | N/T | N/T | normal | N/T |
| D202R | 6.8 | N/T | N/T | N/T | normal | N/T |
| R440AK441A (RK-2A) | 4.2 | N/T | N/T | N/T | normal | N/T |
| R440EK441E (RK-2E) | 11.2 | N/T | N/T | N/T | normal | N/T |
| R6ED333K | 2.6 | N/T | N/T | N/T | **very poor** | N/T |
| K7ED333K | 1.7 | N/T | N/T | N/T | normal | N/T |
| K9ED333K | **0.1** | N/T | N/T | N/T | **very poor** | N/T |
| R6EE337K | 16.4 | N/T | N/T | N/T | normal | N/T |
| K7EE337K | 12.8 | N/T | N/T | N/T | normal | N/T |
| K9EE337K | **6.4** | N/T | N/T | N/T | **very poor** | N/T |
| D202R RK-2E | 61.6 | *yes* | *yes* | no | normal | N/T |
| TelA (1–439) ΔC3 | 8.1 | N/T | N/T | N/T | normal | N/T |
| ΔC3; D202A | 14.2 | N/T | N/T | N/T | normal | N/T |
| ΔC3; D202AE337K | 84.2 | no | no | no | normal | normal |
| ΔC3; D202RE337K | 89.4 | *yes* | *yes* | no | normal | normal |
| ΔN | 0 | no | *yes* | no | no | no |
| ΔN; E337K | 218 | no | no | no | no | normal |
| ΔN; D202RE337K | 156 | *yes* | *yes* | *yes* | no | |
| ΔNΔC3; E337K | 0.8 | N/T | N/T | N/T | **very poor** | **very poor** |
| ΔNΔC3; D202RE337K | 198 | *yes* | *yes* | *yes* | normal | normal |
| N | no | N/T | N/T | N/T | poor | normal |
| N; K9E | no | N/T | N/T | N/T | **very poor** | **no** |

**tel. res.** denotes telomere resolution; **recombin.** denotes recombination; **topo.** denotes topoisomerase activity on a wild type substrate plasmid; *rTel* bind denotes EMSA assays on substrate DNA; N/T denotes not tested. **Bolded** text within the cells indicates hypoactive activities indicative misfolded proteins. *Italicized* text within the cells indicates these inappropriate activities have been unleashed due loss of autoinhibition.

[1] fold-stimulation over wild type TelA measured at 0˚C in the presence of 2 mM $CaCl_2$.

error plot for the model is presented in S2 Fig in S1 File and indicates a low confidence (~30 Å) in the prediction for interface 1 but much higher confidence for interface 2 (~10 Å). We interrogated the two modeled autoinhibitory interfaces by generating and purifying TelA mutants predicted to weaken and/or abolish these interfaces and assessing their telomere resolution activity. The results called to mind the famous aphorism coined by the statistician George Box that "All models are wrong, but some models are useful" [24]. While our mutagenic interrogation of the modeled interfaces did not support the details of the model, the model was, nonetheless, useful in discovering activating mutations that, when combined, led

to a dramatic (50-280-fold) activation of telomere resolution by TelA. For most hyperactive mutants the resulting activation has come at the expense of activating competing reactions that telomere resolvases can promote. Vitally, we have also recovered hyperactive mutants that retain enough autoinhibition **not** to promote the competing reactions of hp telomere fusion or recombination between *rTel* junctions.

Possession of enzymes that promote DNA breakage and rejoining activities is risky as unrepaired strand scissions can lead to genomic instability and cell death. Unsurprisingly, such enzymes are under multiple levels of control/regulation [25–27]. The most fundamental control mechanisms are intrinsic to the enzyme itself, often in the form of autoinhibition [28]. An example from an enzyme class structurally and mechanistically related to the telomere resolvases is the tyrosine recombinase Cre. Cre promotes recombination between short inverted repeat recognition sites called *loxP;* Cre synapses a pair of *loxP* sites together via assembly of a tetramer of Cre [29]. The tetrameric assembly possesses two-fold symmetry that helps define active protomers *vs*. inactive protomers, as Cre executes a pairwise series of DNA cleavage and strand transfer steps, transiting through a HJ intermediate to produce recombinants [30–32]. A key aspect of this two-fold symmetry is the *trans* swapping of the C-terminal helix of Cre between protomers. Cre has been observed in a monomeric state before DNA binding in which the C-terminal helix is stacked in *cis* over the DNA binding cleft and active site of unbound Cre. DNA binding induces a *cis*-to-*trans* switch in the conformation of the C-terminal helix [33]. Our modeling suggested that TelA may also undergo such a *cis*-to-*trans* switch in the conformation of the C-terminal helix between the transition from unbound monomer to DNA bound dimer (Fig 1 and S2 Fig in S1 File). While the model details we tested were not supported by the mutagenic study of the charged residues D202, R440 and K441 the strong stimulation of the reaction by mutants of these residues lends some support to this hypothesis. The specific reason mutation of these residues was activating may have been confounded by the role the D202 residue plays in divalent metal ion responsiveness [16]. Interestingly, the D202R mutation could also be stimulatory due to enhanced DNA binding as the charge reversal at this position is predicted to allow additional DNA contacts with the hairpin turnaround in the crystallographic model with DNA (S9 Fig in S1 File). We were hampered in our ability to test other residues at this interface in positions modeled with higher confidence (<10Å) by the high density of residues in the hairpin binding module important in activity for telomere resolution [12].

A further, much lower confidence (~30Å), autoinhibitory interaction is hypothesized between the N-terminal domain and the catalytic domain (Fig 1, S2 Fig in S1 File and ref. [16]). Our paired charge reversal analysis lent some support for K7 interactions with D333 and potentially with E337 (Fig 4). However, the E337K mutation increases the affinity of TelA for the substrate DNA and, thus, affords stimulation by means independent of a contribution to an autoinhibitory interface (S6 Fig in S1 File). We conclude this because deletion of the N-terminal domain plus incorporation of the E337K mutation that increases affinity for *rTel* DNA led to additional TelA activation rather than the expected neutral effect. Addition of the N-terminal domain in *trans* inhibited telomere resolution confirming the autoinhibitory effect of the N-terminal domain but indicating that multiple or alternative interfaces likely exist beyond the one modeled. The D333K and E337K mutations were introduced digitally into a crystallographic model of TelA complexed with DNA; increased DNA affinity is not predicted by this analysis without some conformational change in TelA as the residues point away from the substrate DNA (S9 Fig in S1 File).

The (ΔN; E337K) mutant has fulfilled the principle prediction of the N-terminal domain's hypothesized role as an autoinhibitory domain of deletion of the domain leading to hyperactivity of the enzyme [28]. The ΔN mutant was activated for competing reactions under certain

conditions but did not show hyperactivity in telomere resolution (ref. [16] and Fig 5B). The lack of hyperactivity was due to the loss of affinity with the *rTel* substrate DNA. However, incorporation of the E337K mutation has reversed this loss of DNA affinity and revealed the activation afforded by loss of the N-terminal domain (>200-fold activation for EDTA and $Ca^{2+}$ conditions). The (ΔN; E337K) mutant will have particularly high utility as an activated version of TelA since it retains a sufficient degree of autoinhibition, presumably by *cis* interactions of the C-terminal helix, not to be activated for the competing reactions of hp telomere fusion and recombination between *rTels* (Figs 6 & 7). Also of great potential utility is the (ΔC3; D202AE337K) mutant that has hyperactivated TelA without activating competing reactions, presumably by continued autoinhibition via the N-terminal domain. Our collection of hyperactive TelA mutants will also be useful for future studies, perhaps by cryo-EM, to characterize the structural basis of autoinhibition and activation of TelA.

## Supporting information

**S1 Raw images.**
(PDF)

**S1 File.**
(PDF)

## Acknowledgments

We would like to thank lab and research cluster members for fruitful discussions.

## Author Contributions

**Conceptualization:** Kerri Kobryn.

**Data curation:** Kerri Kobryn.

**Formal analysis:** Kerri Kobryn.

**Funding acquisition:** Kerri Kobryn.

**Investigation:** Shu Hui Huang, Kayla Abrametz, Siobhan L. McGrath.

**Project administration:** Kerri Kobryn.

**Supervision:** Shu Hui Huang, Kerri Kobryn.

**Writing – original draft:** Kerri Kobryn.

**Writing – review & editing:** Kerri Kobryn.

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
