## [Decision Letter · Decision Letter 0]

2 Jun 2024

PONE-D-24-17544Design and Characterization of Hyperactive Mutants of the *Agrobacterium tumefaciens* Telomere Resolvase, TelA.***PLOS ONE*

Dear Dr. Kobryn,

*Thank you for submitting your manuscript to PLOS ONE. After careful consideration, we feel that it has merit but does not fully meet PLOS ONE’s publication criteria as it currently stands. Therefore, we invite you to submit a revised version of the manuscript that addresses the points raised during the review process.*

Based on the critiques of reviewers and of my independent reading, the following comments need to be addressed:

I concur with Reviewer 1 that a score be presented the best fit AlphaFold model. A poor model might explain the lack of data support. This does not take of course take away from the interesting effects of the mutants just the logical framework of the data presentation.Reviewer 1 also suggests testing in a subset of mutants, in which activity is varied by altering the concentration of the enzyme.  Either conduct such an experiment for the most hyperactive mutant or provide a rebuttal as to why you feel that this is uncessary.Answer all other critiques of both Reviewers.

Based on my reading of the text, I suggest the following additional changes to improve the clarity of the paper.

Page 12, line 9: It is unclear to me whether these events behave in a strictly additive way. To Please provide the expectations of additivity vs the results obtained within the text.Page 13, Line 4. You say that D202 and deltaC-3 are additive. Please explain this outcome. Was this outcome expected?Phenotypic separation: Since you discuss annealing activity and other enzyme characteristics at multiple points within the text, it would be very helpful to provide a summary table of all of the characteristics of each mutant/activity.Redefine hp and rTel when first used in the Results section for clarity.Figure 4, how many replicates of each mutant ws used to derive the mean and standard deviation.

*Please submit your revised manuscript by Jul 17 2024 11:59PM. If you will need more time than this to complete your revisions, please reply to this message or contact the journal office at plosone@plos.org. *

*Please include the following items when submitting your revised manuscript:*

*A rebuttal letter that responds to each point raised by the academic editor and reviewer(s). You should upload this letter as a separate file labeled 'Response to Reviewers'.*

*A marked-up copy of your manuscript that highlights changes made to the original version. You should upload this as a separate file labeled 'Revised Manuscript with Track Changes'.*

*An unmarked version of your revised paper without tracked changes. You should upload this as a separate file labeled 'Manuscript'.*

**

*We look forward to receiving your revised manuscript.*

*Kind regards,*

*Arthur J. Lustig, PhD*

Academic Editor

*PLOS ONE*

*Journal Requirements:*

"KK was supported by Discovery Grants from the Natural Sciences and Engineering Research Council of Canada (NSERC; RGPIN 04382-2017, GRF-0-2006, RGPIN-2024-05101) and by a CoMRAD grant from the University of Saskatchewan's College of Medicine (2022-2023)."

*4. Please review your reference list to ensure that it is complete and correct. If you have cited papers that have been retracted, please include the rationale for doing so in the manuscript text, or remove these references and replace them with relevant current references. Any changes to the reference list should be mentioned in the rebuttal letter that accompanies your revised manuscript. If you need to cite a retracted article, indicate the article’s retracted status in the References list and also include a citation and full reference for the retraction notice.*

Reviewers' comments:

*Reviewer's Responses to Questions*

*

**Comments to the Author**
*

1. Is the manuscript technically sound, and do the data support the conclusions?

*The manuscript must describe a technically sound piece of scientific research with data that supports the conclusions. Experiments must have been conducted rigorously, with appropriate controls, replication, and sample sizes. The conclusions must be drawn appropriately based on the data presented. *

*Reviewer #1: Yes*

*Reviewer #2: Yes*

*2. Has the statistical analysis been performed appropriately and rigorously? *

*Reviewer #1: Yes*

*Reviewer #2: Yes*

*3. Have the authors made all data underlying the findings in their manuscript fully available?*

*The PLOS Data policy requires authors to make all data underlying the findings described in their manuscript fully available without restriction, with rare exception (please refer to the Data Availability Statement in the manuscript PDF file). The data should be provided as part of the manuscript or its supporting information, or deposited to a public repository. For example, in addition to summary statistics, the data points behind means, medians and variance measures should be available. If there are restrictions on publicly sharing data—e.g. participant privacy or use of data from a third party—those must be specified.*

*Reviewer #1: Yes*

*Reviewer #2: Yes*

*4. Is the manuscript presented in an intelligible fashion and written in standard English?*

*PLOS ONE does not copyedit accepted manuscripts, so the language in submitted articles must be clear, correct, and unambiguous. Any typographical or grammatical errors should be corrected at revision, so please note any specific errors here.*

*Reviewer #1: Yes*

*Reviewer #2: Yes*

*5. Review Comments to the Author*

*Please use the space provided to explain your answers to the questions above. You may also include additional comments for the author, including concerns about dual publication, research ethics, or publication ethics. (Please upload your review as an attachment if it exceeds 20,000 characters)*

*Reviewer #1: Huang et al. report the design and characterization of many mutant variants of Agrobacterium tumefaciens Telomere Resolvase, TelA. Despite the authors’ initial intention to relieve autoinhibition suggested by an AI-based structural model of DNA-free TelA, they identified several mutations that confer interesting gain-of-function phenotypes through independent and unknown mechanisms. The obtained mutant collection and biochemical data will be instrumental in further investigating the molecular mechanisms of hairpin telomere resolution or adapting TelA for biotechnology applications. Thus, the work seems to be well-suited for publication in this journal. I have several comments.*

1. The AlphaFold prediction of DNA-unbound TelA served as the template for designing the mutants analyzed in this work. Although the individual domain structures are likely to be accurately predicted, it’s not clear if the domain-domain contacts were predicted with high confidence. The authors should show and discuss error estimates in the prediction, especially the Predicted Aligned Error. Also, the original paper reporting AlphaFold (PMID: 34265844) should be cited.

2. Most hairpin resolution assays were done at 0 deg, which is unusual. Data shown in Fig. 5B (WT vs. delta N) suggest that an enzyme completely inactive at 0 deg can be more active than WT at 30 deg. I don’t recommend repeating everything at 30 deg., but some of the most interesting mutants’ data (e.g., activities of the most active mutant) could be confirmed at 30 deg. It should be possible to slow the reaction by lowering the enzyme concentration, etc.

*3. The authors should discuss the potential mechanisms of hyperactive mutants based on the DNA-bound TelA structure. For instance, are the D202R, D333K, and E337K substitutions predicted to generate additional DNA backbone contacts?*

*Reviewer #2: This manuscript is a mutagenic/biochemical analysis of reaction autoinhibition by the telomere resolvase (TelA) of Agrobacterium tumefaciens. The telomere resolvases are a unique class of DNA metabolizing enzyme that generate covalently closed hairpin ends on replication intermediates of linear replicons in several unrelated bacteria and phages. Although not abundantly found, these enzymes are fascinating as effective solutions for the end replication problem.*

In the current work the authors begin with an AlphaFold model of TelA, including the N-terminal domain which is not resolved in the crystal structure and the absence of DNA which was present in the crystal structure. The model suggests two possible inhibitory interfaces which were then explored through alteration of putative participating residues coupled with in vitro analysis of the purified mutant proteins. The experiments are carefully performed and the data is judiciously interpreted. Although the results do not corroborate the molecular details of the model for the autoinhibitory interfaces, they do however, provide a variety of new information on autoinhibition and in particular residues involved in the process. The study has generated a variety of mutants with increased specific activity in telomere resolution activity, up to 280-fold improved over the wild-type enzyme, a very significant accomplishment. This work lays the ground work for some very interesting mechanistic characterization of the autoinhibition process in future by coupling the current data with future structural analysis of the entire TelA protein +/- DNA using cryo-EM; however, that is beyond the scope of the current work which makes a significant contribution to our knowledge of autoinhibition in TelA.

I have no concerns with the manuscript. The work is carefully executed and interpreted and the manuscript is well-constructed.

Minor point:

*- Fig S7 – I can see the topoisomers, however the “topo” labelling mentioned in the legend is not on the figure.*

*6. PLOS authors have the option to publish the peer review history of their article (what does this mean?). If published, this will include your full peer review and any attached files.*

**

*Reviewer #1: **Yes: **Hideki Aihara*

*Reviewer #2: No*

**

*While revising your submission, please upload your figure files to the Preflight Analysis and Conversion Engine (PACE) digital diagnostic tool, https://pacev2.apexcovantage.com/. PACE helps ensure that figures meet PLOS requirements. To use PACE, you must first register as a user. Registration is free. Then, login and navigate to the UPLOAD tab, where you will find detailed instructions on how to use the tool. If you encounter any issues or have any questions when using PACE, please email PLOS at figures@plos.org. Please note that Supporting Information files do not need this step.*

---

## [Author Response · Author response to Decision Letter 0]

4 Jul 2024

Dear Dr. Lustig,

 We thank you and the two reviewers for their expert evaluation of our manuscript ‘Design and Characterization of Hyperactive Mutants of the Agrobacterium tumefaciens Telomere Resolvase, TelA.’ (ONE-D-24-17544). We feel that addressing the reviewer comments/concerns has markedly strengthened the revised manuscript. In brief, we have added a Supplementary Figure detailing the predicted aligned error of the model used in the paper (S2 Fig). A second Supplementary Figure (S9 Fig) details the results of digital mutagenesis to assess introduction of polar contacts with D202R, D333K and E337K mutations with DNA potentially caused by these charge reversal mutations. We have also added an additional data figure in the main text of the paper (Figure 8) wherein we test the two most useful hyperactive mutants against wild type TelA at the standard reaction temperature of 30oC (instead of 0oC) using a low protein concentration to slow reactions for ease of measuring initial rates (3 nM as opposed to 74 nM). The mutants chosen for analysis were hyperactivated (>50-fold) but retained enough autoinhibition to suppress competing reactions. These changes have necessitated the renumbering of supplementary figure references and has lengthened, somewhat, the manuscript to incorporate discussion of this new figure and also discussion of the predicted aligned error results. What follows below is a point by point response to the reviewer/editor comments.

Editor comments:

Based on the critiques of reviewers and of my independent reading, the following comments need to be addressed:

1. I concur with Reviewer 1 that a score be presented the best fit AlphaFold model. A poor model might explain the lack of data support. This does not take of course take away from the interesting effects of the mutants just the logical framework of the data presentation.

We have prepared a new supplementary Figure with the Predicted Aligned Error (S2 Fig). This has necessitated the renumbering of the SFig referents throughout text. We have discussed the error estimates in the results and discussion sections. The take home is that the predicted aligned error estimates reveal that the model is low confidence for interface 1 and much higher confidence for interface 2. Nonetheless, testing both predicted interfaces proved a useful exercise for uncovering activating mutations. 

3. Reviewer 1 also suggests testing in a subset of mutants, in which activity is varied by altering the concentration of the enzyme. Either conduct such an experiment for the most hyperactive mutant or provide a rebuttal as to why you feel that this is uncessary.

This was an excellent suggestion since the specific ranking of activities could be a read out of general activation but also, potentially, a read out of the effect of a mutation specifically on the cold-sensitivity of telomere resolution. We repeated telomere resolution assays for WT, (�C3; D202AE337K) and (�N; E337K) at 30oC at low protein concentration (3 nM). These two mutants were selected since they do not suffer from activation of inappropriate activities due to their hyperactivation. This analysis has become Fig. 8. The results indicate that activation is also seen at 30oC for these two mutants but the relative ranking of their activities has changed indicating that there were differential effects on the cold-sensitivity of the telomere resolution.

Based on my reading of the text, I suggest the following additional changes to improve the clarity of the paper.

Page 12, line 9: It is unclear to me whether these events behave in a strictly additive way. To Please provide the expectations of additivity vs the results obtained within the text.

What is meant by additive is now spelled out and a counter example of non-additive behaviour of a double mutant is provided if the mutations affect a common autoinhibitory interface. It essentially boils down to something akin to an in vitro epistasis analysis.

4. Page 13, Line 4. You say that D202 and deltaC-3 are additive. Please explain this outcome. Was this outcome expected? 

We have clarified this section by indicating what our expectations when combining the D202 mutants with the delta-C3 mutation. The increased activation observed was unexpected and foreshadowed the result of the more rigorous paired charge reversal results documented later in the paper that these residues are unlikely to interact to help form the modeled interface.

5. Phenotypic separation: Since you discuss annealing activity and other enzyme characteristics at multiple points within the text, it would be very helpful to provide a summary table of all of the characteristics of each mutant/activity.

Such a mass of different assay results applied to numerous mutants has now been made more accessible, at a glance, by creation of Table 1 for the main body of the paper. 

6. Redefine hp and rTel when first used in the Results section for clarity.

Completed as requested to improve clarity of presentation of the results.

7. Figure 4, how many replicates of each mutant ws used to derive the mean and standard deviation.

We have changed all figure legends to indicate that the graphed results show the mean and standard deviation of 3 independent replicates excepting the noted mutants that were assayed with 6 replicates. This affords greater clarity.

Reviewer #1 comments:

1) The AlphaFold prediction of DNA-unbound TelA served as the template for designing the mutants analyzed in this work. Although the individual domain structures are likely to be accurately predicted, it’s not clear if the domain-domain contacts were predicted with high confidence. The authors should show and discuss error estimates in the prediction, especially the Predicted Aligned Error.

We have prepared a new supplementary Figure with the Predicted Aligned Error (S2 Fig). This has necessitated the renumbering of the SFig referents throughout text. We have discussed the error estimates in the results and discussion sections. The take home is that the predicted aligned error estimates reveal that the model is low confidence for interface 1 and much higher confidence for interface 2. Nonetheless, testing both predicted interfaces proved a useful exercise for uncovering activating mutations. 

Also, the original paper reporting AlphaFold (PMID: 34265844) should be cited.

The relevant Jumper et al. reference was in the results text but has also been added to the legend for Figure 1.

2) Most hairpin resolution assays were done at 0 deg, which is unusual. Data shown in Fig. 5B (WT vs. delta N) suggest that an enzyme completely inactive at 0 deg can be more active than WT at 30 deg. I don’t recommend repeating everything at 30 deg., but some of the most interesting mutants’ data (e.g., activities of the most active mutant) could be confirmed at 30 deg. It should be possible to slow the reaction by lowering the enzyme concentration, etc.

This was an excellent suggestion since the specific ranking of activities could be a read out of general activation but also, potentially, a read out of the effect of a mutation on the cold-sensitivity of telomere resolution. We repeated telomere resolution assays for WT, (�C3; D202AE337K) and (�N; E337K) at 30oC at low protein concentration (3 nM). These two mutants were selected since they do not suffer from activation of inappropriate activities due to their hyperactivation. This analysis has become Fig. 8. The results indicate that activation is also seen at 30oC for these two mutants but the relative ranking of their activities has changed indicating that there were differential effects on the cold-sensitivity of the telomere resolution.

3) The authors should discuss the potential mechanisms of hyperactive mutants based on the DNA-bound TelA structure. For instance, are the D202R, D333K, and E337K substitutions predicted to generate additional DNA backbone contacts?

We have performed and documented this ‘digital mutagenesis’ using an available TelA co-crystal structural model. This is now S9 Fig. The results show that D202R makes contacts with the DNA absent in the wild type but that D333K and E337K do not unless a modest conformational change occurs due to these mutations.

Reviewer #2 comments:

1) Fig S7 – I can see the topoisomers, however the “topo” labelling mentioned in the legend is not on the figure.

This supplementary figure is now S8 Fig and has the ladder of topoisomers bracketed and labeled as topo as noted in the figure legend. We thank the reviewer for catching this oversight. 

We thank the reviewers for helping to strengthen the revised manuscript.

Best regards, 

Kerri Kobryn, PhD

Associate Professor

Dept of Biochemistry, Microbiology & Immunology

College of Medicine

University of Saskatchewan

kerri.kobryn@usask.ca

---

## [Decision Letter · Decision Letter 1]

9 Jul 2024

Design and Characterization of Hyperactive Mutants of the *Agrobacterium tumefaciens* Telomere Resolvase, TelA.**

*PONE-D-24-17544R1*

*Dear Dr. Kobryn,*

*We’re pleased to inform you that your manuscript has been judged scientifically suitable for publication and will be formally accepted for publication once it meets all outstanding technical requirements.*

*Within one week, you’ll receive an e-mail detailing the required amendments. When these have been addressed, you’ll receive a formal acceptance letter and your manuscript will be scheduled for publication.*

*An invoice will be generated when your article is formally accepted. Please note, if your institution has a publishing partnership with PLOS and your article meets the relevant criteria, all or part of your publication costs will be covered. Please make sure your user information is up-to-date by logging into Editorial Manager at Editorial Manager® and clicking the ‘Update My Information' link at the top of the page. If you have any questions relating to publication charges, please contact our Author Billing department directly at authorbilling@plos.org.*

*If your institution or institutions have a press office, please notify them about your upcoming paper to help maximize its impact. If they’ll be preparing press materials, please inform our press team as soon as possible -- no later than 48 hours after receiving the formal acceptance. Your manuscript will remain under strict press embargo until 2 pm Eastern Time on the date of publication. For more information, please contact onepress@plos.org.*

*Kind regards,*

*Arthur J. Lustig, PhD*

Academic Editor

*PLOS ONE*

* *

*Additional Editor Comments (optional):*

* *

*Reviewers' comments:*

*Reviewer's Responses to Questions*

*

**Comments to the Author**
*

*1. If the authors have adequately addressed your comments raised in a previous round of review and you feel that this manuscript is now acceptable for publication, you may indicate that here to bypass the “Comments to the Author” section, enter your conflict of interest statement in the “Confidential to Editor” section, and submit your "Accept" recommendation.*

*Reviewer #1: All comments have been addressed*

*2. Is the manuscript technically sound, and do the data support the conclusions?*

*The manuscript must describe a technically sound piece of scientific research with data that supports the conclusions. Experiments must have been conducted rigorously, with appropriate controls, replication, and sample sizes. The conclusions must be drawn appropriately based on the data presented. *

*Reviewer #1: Yes*

*3. Has the statistical analysis been performed appropriately and rigorously? *

*Reviewer #1: Yes*

*4. Have the authors made all data underlying the findings in their manuscript fully available?*

*The PLOS Data policy requires authors to make all data underlying the findings described in their manuscript fully available without restriction, with rare exception (please refer to the Data Availability Statement in the manuscript PDF file). The data should be provided as part of the manuscript or its supporting information, or deposited to a public repository. For example, in addition to summary statistics, the data points behind means, medians and variance measures should be available. If there are restrictions on publicly sharing data—e.g. participant privacy or use of data from a third party—those must be specified.*

*Reviewer #1: Yes*

*5. Is the manuscript presented in an intelligible fashion and written in standard English?*

*PLOS ONE does not copyedit accepted manuscripts, so the language in submitted articles must be clear, correct, and unambiguous. Any typographical or grammatical errors should be corrected at revision, so please note any specific errors here.*

*Reviewer #1: Yes*

*6. Review Comments to the Author*

*Please use the space provided to explain your answers to the questions above. You may also include additional comments for the author, including concerns about dual publication, research ethics, or publication ethics. (Please upload your review as an attachment if it exceeds 20,000 characters)*

*Reviewer #1: (No Response)*

*7. PLOS authors have the option to publish the peer review history of their article (what does this mean?). If published, this will include your full peer review and any attached files.*

**

*Reviewer #1: **Yes: **Hideki Aihara*

---

## [Editor Report · Acceptance letter]

16 Jul 2024

PONE-D-24-17544R1 

PLOS ONE

Dear Dr. Kobryn, 

I'm pleased to inform you that your manuscript has been deemed suitable for publication in PLOS ONE. Congratulations! Your manuscript is now being handed over to our production team.

Kind regards, 

on behalf of

Dr. Arthur J. Lustig 

Academic Editor

PLOS ONE